# Poly(methyl methacrylate) Bone Cement Composite Can Be Refilled with Antibiotics after Implantation in Femur or Soft Tissue

**DOI:** 10.3390/jfb12010008

**Published:** 2021-01-26

**Authors:** Erika L. Cyphert, Ningjing Zhang, Dylan W. Marques, Greg D. Learn, Fang Zhang, Horst A. von Recum

**Affiliations:** Department of Biomedical Engineering, Case Western Reserve University, 10900 Euclid Avenue, Cleveland, OH 44106, USA; elc50@case.edu (E.L.C.); nxz137@case.edu (N.Z.); dwm87@case.edu (D.W.M.); gdl25@case.edu (G.D.L.); fxz118@case.edu (F.Z.)

**Keywords:** PMMA, antibiotic, bone cement, refill, femur

## Abstract

While periprosthetic joint infections (PJIs) result in a small percentage of patients following arthroplasties, they are challenging to treat if they spread into bone and soft tissue. Treatment involves delivering antibiotics using poly(methyl methacrylate) (PMMA) bone cement. However, antibiotic release is insufficient for prolonged infections. Previous work demonstrated efficacy of incorporating insoluble cyclodextrin (CD) microparticles into PMMA to improve antibiotic release and allow for post-implantation drug refilling to occur in a tissue-mimicking model. To simulate how antibiotic refilling may be possible in more physiologically relevant models, this work investigated development of bone and muscle refilling models. The bone refilling model involved embedding PMMA-CD into rabbit femur and administering antibiotic via intraosseous infusion. Muscle tissue refilling model involved implanting PMMA-CD beads in bovine muscle tissue and administering antibiotic via tissue injection. Duration of antimicrobial activity of refilled PMMA-CD was evaluated. PMMA-CD composite in bone and muscle tissue models was capable of being refilled with antibiotics and resulted in prolonged antimicrobial activity. PMMA-CD provided sustained and on-demand antimicrobial activity without removal of implant if infection develops. Intraosseous infusion appeared to be a viable technique to enable refilling of PMMA-CD after implantation in bone, reporting for the first time the ability to refill PMMA in bone.

## 1. Introduction

Periprosthetic joint infections (PJIs) can be a devastating complication resulting from some arthroplasties and can involve infection of both the soft tissue surrounding the prosthetic implant as well as the inside of the bone in severe cases [1]. To mitigate the development of and to treat PJIs, antibiotic-laden poly(methyl methacrylate) (PMMA) bone cement is often utilized where it is either directly implanted in the bone (i.e., arthroplasty fixation) or in the form of beads that are embedded into the surrounding infected soft tissue to locally deliver antibiotics [2,3,4,5,6,7,8]. When antibiotic-laden PMMA is directly implanted into the bone, there is a unique challenge to administer additional antibiotic doses into the implanted PMMA if the infection is prolonged. As more drug cannot be added to the PMMA after it is implanted in the bone, the duration of antimicrobial activity is limited based upon the initial amount of antibiotic added to the PMMA upon fabrication [9,10]. When antibiotic-laden PMMA beads are implanted and packed into soft tissue near the infected surgical site in arthroplasty revision procedures [11,12,13], the duration of antimicrobial activity of antibiotic-laden PMMA beads can be limited if a bacterial biofilm or resistant bacteria adhere to the surface of the beads [12,13]. Specifically, if antibiotic is released from the PMMA beads at sub-therapeutic levels, bacteria can adhere to and form a thick biofilm matrix that is impenetrable to many antibiotics, rendering the treatment unsuccessful [13].

Traditionally, to prepare antibiotic-laden PMMA, antibiotics such as aminoglycosides and glycopeptides are often incorporated into the PMMA during polymerization [14]. However, when antibiotics are added to PMMA in this manner, it often results in an insufficient release of drug. Typically, a “burst” release is observed initially where a substantial amount of drug is released within a very short period of time and is followed by a prolonged period of antibiotic release at sub-therapeutic levels [15]. Oftentimes, much of the drug added initially can remain entrapped in the PMMA permanently with one study showing upwards of 85% of drug remaining trapped over an extended period of time [16]. In conjunction with the antibiotic-filled PMMA, additional antibiotics may be administered either intravenously or intraosseously to treat PJIs and orthopedic infections in an effort to supplement the poor antibiotic release kinetics from PMMA [17,18,19]. Intraosseous (IO) infusion offers a unique advantage over traditional intravenous administration as it can provide a higher local tissue antibiotic concentration while reducing off-target systemic effects [17]. IO infusion is generally used in clinical scenarios in which venous access is compromised and involves placing a specialized needle (16 or 18 gauge, 2.5–3 cm long) directly into the medullary cavity in the bone marrow (distal femur or proximal tibia) and remains in place as a port (up to 72 h) to provide direct venous access for therapeutics [20,21].

In an effort to improve upon the antibiotic release kinetics and to enable post-implantation antibiotic refilling to occur, a PMMA composite containing polymerized cyclodextrin (CD) microparticles was previously developed [5,8]. Cyclodextrin contains between six and eight glucose rings and has an inner “pocket” that is slightly hydrophobic, able to bind and release drugs based upon the strength of their binding energy with CD [22,23,24]. Through this previous work we demonstrated that when CD microparticles were added to PMMA, the PMMA was able to be refilled with antibiotics following implantation in an agarose-based tissue phantom and exhibited additional windows of prolonged antimicrobial therapy that was not possible with PMMA on its own [5,8]. While much success has been observed using the agarose-based refilling model, it was not the most representative of bone or muscle tissue. To better recapitulate what PMMA-CD composite refilling may look like to treat two aspects of PJIs where (1) PMMA is embedded in bone for treatment of osteomyelitis and (2) PMMA beads are embedded in soft muscle tissue for treatment of infections surrounding the bone, both bone and muscle tissue antibiotic refilling models were developed in this work.

To simulate how antibiotic refilling of PMMA-CD composite embedded in femur could reasonably occur, a refilling model was developed in which PMMA-CD composite was embedded in the bone and IO infusion was used to provide direct access to the bone marrow to administer antibiotics and allow for refilling of embedded PMMA-CD. The antibiotic refilling ability of PMMA with and without CD microparticles embedded in femur was evaluated through post-refilling analysis of the depth of antibiotic diffusion into PMMA-CD composite and the resultant duration of antimicrobial activity. To translate the refilling model for the application of PMMA beads to treat muscle tissue PJIs, PMMA beads with and without CD microparticles were implanted in bovine muscle tissue and the tissue was directly injected with a bolus of antibiotic. Extent of refilling capacity of PMMA-CD composite beads was evaluated through administration of iterative antibiotic doses and subsequently investigating the resultant duration of antimicrobial activity of refilled PMMA-CD beads. To explore the possibility of refilling PMMA-CD composite beads through muscle tissue if they are initially covered in bacterial biofilm, a modified soft-tissue refilling model was developed where biofilms were formed on composite beads prior to implantation in the tissue. Duration of antimicrobial activity of refilled PMMA-CD composite beads with biofilm was evaluated and the remaining colony-forming unit counts on the bead and in the surrounding tissue were quantified after antibiotic treatment. Implementation of the proof-of-concept bone and muscle tissue refilling models developed in this work could provide insight into novel treatment options to improve outcomes for patients with chronic or prolonged PJIs.

## 2. Materials and Methods

### 2.1. Materials

Lightly epichlorohydrin cross-linked β-CD pre-polymer (116 kDa molecular weight) was purchased from CycloLabs (Budapest, Hungary). Ethylene glycol diglycidyl ether was purchased from Polysciences Inc. (Warrington, PA, USA). Rifampin (RMP) was purchased from Research Products International (Mt. Prospect, IL, USA). Simplex^®^ HV (high viscosity) radiopaque bone cement (Stryker Orthopaedics, Mahwah, NJ, USA) and Cook^®^ Medical Intraosseous (IO) infusion needles (16 gauge, 4 cm) (Bloomington, IN, USA) were purchased from eSutures (Mokena, IL, USA). New Zealand white rabbit knee joint tissue was graciously donated by the Animal Resource Center (ARC) at Case Western Reserve University (CWRU, Cleveland, OH, USA). Bovine abdominal muscle tissue was purchased from a local butcher (Cleveland, OH, USA). Green fluorescent protein (GFP)-labeled *Staphylococcus aureus* bacteria (Seattle 1945 strain, ATCC, Manassas, VA, USA) was kindly provided by Dr. Edward Greenfield (CWRU, Cleveland, OH, USA). All other reagents were purchased in the highest grade possible from Fisher Scientific (Pittsburgh, PA, USA).

### 2.2. β-CD Microparticle Synthesis

Insoluble cyclodextrin (β-CD) microparticles were synthesized according to a previously published methodology [5,8,25]. Specifically, 1 g of epichlorohydrin cross-linked β-CD was dissolved in 4 mL of 0.2 M potassium hydroxide and cross-linked with ethylene glycol diglycidyl ether (1.6 mL). The mixture was heated at 60 °C in a mixture of light mineral oil (50 mL) and 750 μL surfactant mix (24% Tween 85, 76% Span 85) and stirred for 4 h. CD microparticles were repeatedly washed and centrifuged with the following solvents (2× each, 35 mL of each): light mineral oil, hexanes, acetone, and MilliQ water. CD microparticles were lyophilized and stored in a desiccator until further use. Average approximate CD microparticle diameter was previously determined to be ~200 μm via scanning electron microscopy [25]. Additionally, CD microparticles have been shown to be robust and withstand degradation after prolonged implantation in vivo [25].

### 2.3. PMMA-CD Composite Preparation

For both bone and muscle tissue refilling models, PMMA was prepared using a similar methodology, according to manufacturer instructions. Specifically, 2-g samples of Simplex^®^ HV surgical grade bone cement powder was mixed with 1 mL of methyl methacrylate monomer until a soft dough formed. For PMMA samples containing CD microparticles, 15 wt% CD microparticles (wt/wt) were added to each batch of PMMA prior to addition of liquid monomer and mixed until homogeneous. For the bone refilling model PMMA samples, the soft dough was directly finger-pressed and cured in the bone (displacing bone marrow) and had dimensions of approximately 6 mm tall and 4 mm diameter. For the muscle tissue refilling model PMMA samples, the soft dough was pressed into a thin layer (~2 mm thick) and small beads were punched out (6 mm diameter) and allowed to cure at room temperature. Properties of PMMA-CD composites have previously been characterized extensively including the ultimate compressive strength (~65 MPa) [26], porosity (pore volume fraction = ~1.5–2%) [26], and antibiotic release kinetics [5,8].

### 2.4. Femur PMMA-CD Composite Antibiotic Refilling Model

To simulate how we envisioned PMMA-CD composite antibiotic refilling in the scenario of treating arthroplasty PJIs, we set out to develop a PMMA-CD refilling model of the femur. Rabbit femurs were selected as an ideal ex vivo model, as they are frequently used in orthopedic in vivo models and rabbit bone mineral density has similarities to humans [27,28]. Femurs were harvested from New Zealand white female adult rabbits and the femoral head epiphysis and metaphysis and patellar epiphysis and metaphysis were removed. The remaining diaphysis shaft was measured and cut in half, yielding two femoral samples from one bone. Bone marrow remained intact throughout. The innermost cut end of each bone was filled (6 mm deep) with PMMA-CD composite (either with or without 15 wt% CD microparticles) and allowed to harden. The opposite end of the bone was capped with a thin layer of PMMA (without CD microparticles). An IO infusion needle (16 gauge, 4 cm) was used to drill a hole into the femur 11 mm away from the end of the bone filled with PMMA-CD composite and remained in place to serve as a port to directly inject antibiotic into the medullary cavity. The IO infusion needle was secured in place. The femur section containing the IO infusion needle was placed on top of 5 mL of solidified agarose (0.075% wt/vol) in a six-well plate and covered with an additional 8 mL of agarose to fully cove the bone sample [29]. Once the agarose solidified, 100 μL of RMP (12 mg/mL in methanol) was injected directly into the IO needle and into the bone marrow of the femur. The plate was covered and placed in a 37 °C incubator for 48 h. After incubation, bone segments were removed from the agarose, the IO infusion needle was removed, and the bone was sliced in half vertically in order to visualize the depth of refilling of RMP into the embedded PMMA-CD composite. Each condition was carried out in quadruplicate, consisting of both halves of the right and left femur of a single rabbit.

### 2.5. Analysis of Bone Refilling PMMA-CD Composite—Depth of RMP Refilling and Duration of Antimicrobial Activity

In an effort to evaluate the extent of antibiotic refilling possible with PMMA-CD composite when it was embedded in the femur, refilled PMMA-CD composite was explanted from the femur, sliced in half vertically, and analyzed using two metrics. First, stereomicroscope images were collected of the interior cut face of the PMMA-CD composite while it was still embedded in the bone and the front and back of the explanted PMMA-CD composite (with and without 15 wt% CD microparticles) to provide direct visualization of refilling. Depth or distance of diffusion of RMP refilling into PMMA-CD was directly quantified from the microscope images using ImageJ (2018) where up to eight measurements were collected on each half of the sample measured from the edge of the sample until the end of the visible “orange/red” region on the interior of PMMA-CD (based upon visual inspection) and averaged.

Then, to determine the duration of antimicrobial activity possible from refilled PMMA-CD composite, explanted PMMA-CD samples were evaluated in a persistence zone of inhibition study against *Staphylococcus aureus* (*S. aureus*) bacteria [5,8]. Specifically, 70 μL of *S. aureus* culture was spread evenly on Trypticase soy agar petri dishes and the interior cut RMP refilled face of the PMMA-CD composite was placed face down on the surface of the agar. Dishes were incubated overnight at 37 °C and the following day the zone of inhibition was measured from the edge of the bacterial clearance to the edge of the PMMA-CD sample using calipers (four measurements per sample). PMMA-CD samples were transferred to a fresh *S. aureus* dish and this process was repeated until the zone of inhibition was no longer visible.

### 2.6. Muscle Tissue PMMA-CD Composite Antibiotic Refilling Model

In an effort to recapitulate the scenario in which antibiotic-filled PMMA-CD composite beads are used to treat PJIs, a PMMA-CD antibiotic refilling model was developed to mimic RMP refilling through bovine muscle. Small sections of muscle tissue (2 cm × 2 cm × 1 cm) were prepared, a small horizontal incision was made in the side (1 cm thick) of the tissue, and a PMMA-CD composite bead (with or without 15 wt% CD microparticles) was implanted. A small amount of cyanoacrylate glue was used to seal off the incision. The tissue containing the PMMA-CD composite bead was then embedded in 0.075% wt/vol agarose (13 mL total) in a six-well plate, as described previously [29]. Once the agarose set, two 100-μL injections of RMP (12 mg/mL dissolved in methanol; 200 μL total) were injected directly into the tissue at different locations, approximately 6 mm away from the implanted PMMA-CD composite bead. The plate was then covered and incubated at 37 °C for 48 h. Following 48 h of incubation, tissue was either removed from the agarose and the PMMA-CD beads were explanted and analyzed (for one cycle of refilling) or an additional 200-μL bolus of RMP (12 mg/mL in methanol) was injected into the tissue (two 100-μL injections directly into tissue) and incubated for an additional 48 h (for two cycles of refilling or repeated twice for three cycles of refilling). Each condition was completed in triplicate.

### 2.7. Analysis of Antibiotic Refilling PMMA-CD Composite Beads in Muscle Tissue—Stereomicroscope Images, Duration of Antimicrobial Activity

Following 1–3 cycles of refilling, PMMA-CD composite beads were explanted from the muscle tissue and analyzed using two methods to determine the extent of antibiotic refilling possible in soft tissue. First, images of both the front and back of each PMMA-CD composite bead (with and without 15 wt% CD microparticles) were collected using a stereomicroscope to provide a direct visualization of the antibiotic refilling. Then, the duration of antimicrobial activity possible from each refilled PMMA-CD bead was determined through a persistence zone of inhibition study against *S. aureus* bacteria, according to a previously described methodology [5,8].

### 2.8. Muscle Tissue Infection PMMA-CD Composite Antibiotic Refilling Model

To evaluate the possibility of allowing for antibiotic refilling to occur in soft tissue in the scenario in which a bacterial biofilm has formed on the surface of the implanted PMMA-CD composite bead, a modified muscle tissue infection antibiotic refilling model was developed. Prior to implantation in the muscle tissue sample (2 cm × 2 cm × 1 cm), PMMA-CD composite beads (with and without 15 wt% CD microparticles) were submerged in 1.5 mL of *S. aureus* culture and incubated for 48 h at 37 °C. PMMA-CD beads were then lightly dried off and embedded into the muscle tissue sample and the tissue was held together with cyanoacrylate glue and embedded in 0.075% wt/vol agarose (5 mL bottom, 8 mL top) in a six-well plate and allowed to solidify. Tissue was incubated for 24 h at 37 °C to allow for the infection to further develop. After this time, two 100-μL injections of RMP (12 mg/mL in methanol; 200 μL total) were injected directly into the tissue at different locations ~6 mm away from the implanted PMMA-CD composite bead and the plate was covered and incubated at 37 °C for 48 h. As a control, a subset of tissues containing PMMA-CD composite (with and without 15 wt% CD microparticles) was not injected with any RMP to provide growth control counts of the *S. aureus* colony-forming units (CFUs) without the presence of drug (injection of 200 μL blank methanol control after 24 h incubation, 72 h incubation total prior to explanting). After 48 h of incubation with drug (or blank methanol control), tissue was removed from the agarose and the PMMA-CD composite bead was explanted and analyzed.

### 2.9. Analysis of Antibiotic Refilling PMMA-CD Composite with Biofilm—Extent of Infection Remaining, Duration of Antimicrobial Activity

To evaluate the extent of antibiotic refilling possible in the presence of a bacterial biofilm on the surface of the PMMA-CD composite bead through muscle tissue, the explanted antibiotic refilled PMMA-CD beads were analyzed using two metrics. First, the extent of infection remaining in the tissue and on the surface of the PMMA-CD composite beads after treatment with RMP or methanol was analyzed through quantification of CFU counts. All the tissue surrounding the PMMA-CD composite beads was homogenized in 15 mL of 2× Trypticase soy broth. PMMA-CD composite beads were sonicated for 30 min in 3 mL of 2× Trypticase soy broth. For both solutions (from tissue and PMMA-CD composite beads), 70-μL aliquots of solution were plated out on Trypticase soy agar plates and incubated overnight at 37 °C. CFU counts were completed using ImageJ (2018). Extent of infection remaining after treatment with antibiotic was compared relative to the CFU count in control samples without any antibiotic. Separately, to analyze the duration of antimicrobial activity of PMMA-CD composite beads refilled with RMP through biofilm, refilled beads were explanted from the tissue and evaluated in a persistence zone of inhibition study against *S. aureus* [5,8].

### 2.10. Statistical Analysis

All data were reported as the mean and standard deviation of a minimum of 3–4 samples. Statistical analyses were carried out in Microsoft Excel 2016. Two-tailed Student’s *t*-tests with unequal variances were conducted for depth of antibiotic diffusion (*n* = 4), persistence zone of inhibition (*n* = 3), and CFU counts (*n* = 3). The *p*-values of less than 0.05 were considered to be statistically significant (**p* < 0.05, ***p* < 0.01).

## 3. Results

### 3.1. Femur PMMA-CD Composite Antibiotic Refilling Model

Figure 1 depicts a schematic of the general setup of the femur PMMA-CD composite refilling model developed. Once the femur model was prepared, a bolus of RMP was injected into the IO port and the bone segment was incubated for 48 h. Figure 2 depicts an image of the prepared femur segment with PMMA-CD composite embedded, insertion of the IO needle, and the interior of the sliced femur segment after 48 h of refilling with RMP. Refilling of PMMA-CD composite embedded in the femur model was compared to refilling of PMMA without CD microparticles, as a control.

Figure 3 depicts representative images of the interiors of the PMMA-CD composite (with and without 15 wt% CD microparticles) that was embedded in the femur model and refilled with RMP for 48 h. Results from Figure 3 quantitatively confirmed that refilling of PMMA-CD composites was possible using the IO infusion technique as indicated by the presence of the red/orange color along the periphery of the interior of the sliced PMMA-CD. Specifically, RMP had a red/orange color and the presence of this color along the periphery of the PMMA-CD was indicative of RMP refilling. It was important to note the lack of the “orange” periphery on the plain PMMA samples (without CD microparticles), highlighting the necessity of the inclusion of CD microparticles into the PMMA to enable antibiotic refilling and demonstrating that the “orange” periphery was not due to staining from blood or bone marrow in the femur model, but rather drug.

To quantify the relative amount of refilling possible in samples with and without CD microparticles using the IO infusion technique, stereomicroscope images were collected of the interior of all embedded PMMA-CD samples and ImageJ was used to measure the depth or distance of diffusion of RMP into PMMA-CD. This measurement provided insight regarding how much PMMA-CD material (i.e., thickness) was feasibly used in drug refilling and delivery functions. Figure 4 provides quantification of the depth or distance of diffusion of RMP refilling into PMMA-CD composite with and without CD microparticles and depicts how representative measurements were completed. Up to eight measurements were completed on each sample from the edge to the interior where the “orange” color ended, based upon visual inspection. Measurements were completed and averaged on both halves from four independent samples of each condition. PMMA-CD composite samples containing 15 wt% CD microparticles refilled RMP an average depth of 295 ± 50 μm, whereas samples without CD microparticles refilled RMP a comparably negligible amount (60 ± 8 μm) (*p* = 1.5 × 10^−6^).

To determine the duration of time in which RMP refilled PMMA-CD composites (with and without CD microparticles) were able to provide therapeutically relevant antimicrobial activity, refilled PMMA-CD samples were evaluated in a persistence zone of inhibition study against *S. aureus*. PMMA-CD samples were sliced in half and placed interior-side down on agar plates seeded with *S. aureus*. Figure 5 displays how the size of the zone of inhibition changed when PMMA-CD refilled samples were challenged to a fresh lawn of bacteria every day. The use of IO infusion technique resulted in ~14 days of additional antimicrobial activity after PMMA-CD composite was implanted in bone. More specifically, Figure 5 demonstrates that PMMA containing 15 wt% CD microparticles was able to clear *S. aureus* for >14 days, whereas PMMA without CD microparticles (plain) was only able to clear *S. aureus* for <7 days. Over the first six days of the study, PMMA with 15 wt% CD microparticles had a significantly larger-sized zone of inhibition than the control, plain PMMA (*p* < 0.05; except after one day where *p* > 0.05).

### 3.2. Muscle Tissue PMMA-CD Composite Antibiotic Refilling Model

Figure 6 depicts a schematic of the setup of the muscle tissue PMMA-CD composite antibiotic refilling and infection antibiotic refilling models where (1) the ability of PMMA-CD beads containing 15 wt% CD microparticles to be refilled with RMP through muscle tissue and (2) the ability of the same beads to be refilled with antibiotics in the presence of a bacterial biofilm were analyzed. The muscle tissue refilling model (without a biofilm) was carried out over three consecutive iterations of antibiotic refilling to probe the maximum amount of RMP that could be refilled into the implanted PMMA-CD beads and to evaluate if refilling could be repeatedly performed in the model. This experiment functioned to simulate the scenario in which the patient may require multiple doses of antibiotic for persistent soft tissue PJIs. Figure 7 provides a visual representation of the setup of the implantation of the PMMA-CD bead in the tissue in agarose, injection of RMP, and incubation in tissue.

PMMA-CD beads with and without 15 wt% CD microparticles were implanted in the muscle tissue and refilled with either 1, 2, or 3 consecutive bolus injections of RMP. Following incubation with RMP over 48 h, PMMA-CD beads were explanted from the tissue and imaged using stereomicroscopy. Figure 8 depicts the images of the explanted PMMA-CD beads after 1–3 cycles of RMP refilling. Images in Figure 8 qualitatively reveal that, as additional injections of RMP were subsequently administered in the muscle tissue model, the amount of RMP refilled in the PMMA-CD beads with 15 wt% CD microparticles increased, as indicated by the increased red/orange color. PMMA-CD initially had a white/opaque color; therefore, when exposed to the red/orange RMP, the color change of the PMMA-CD indicated RMP refilling. Without CD microparticles (i.e., plain PMMA), the PMMA demonstrated very little to no RMP refilling across three cycles.

To evaluate the duration of antimicrobial activity possible from PMMA-CD samples refilled over three cycles, a persistence zone of inhibition study was carried out against *S. aureus*. Figure 9 displays the results of the persistence zone of inhibition study for PMMA-CD beads refilled over 1–3 cycles with RMP. Results from the persistence zone of inhibition study (Figure 9) demonstrated that through repetitive refilling in muscle tissue, PMMA-CD composite beads had the ability to provide additional windows of antimicrobial therapy (~25–40+ days), relative to plain PMMA. More specifically, Figure 9 demonstrates that plain PMMA beads (without CD microparticles) were only able to clear bacteria for ~8 days independent of the number of RMP refilling cycles. In contrast, PMMA-CD beads were able to clear bacteria for ~25 days (one or two cycles of RMP refilling) or >40 days (three cycles of RMP refilling). Generally, the size of the zone of inhibition for plain PMMA beads was significantly smaller than that of PMMA-CD composite refilled with RMP for the same number of cycles (*p* < 0.05).

### 3.3. Muscle Tissue Infection PMMA-CD Composite Antibiotic Refilling Model

When biomaterials remain implanted in a patient over a prolonged period of time, they pose the risk of forming bacterial biofilms on their surface, particularly if they are being used in antimicrobial applications [30,31,32]. As a result, we were interested in exploring how antibiotic refilling of PMMA-CD beads through muscle tissue may or may not be impacted if a bacterial biofilm has formed on the surface of the PMMA-CD. *S. aureus* biofilms were statically formed on the surface of the PMMA-CD beads over 48 h prior to implantation in muscle tissue. Once implanted, the tissue was incubated for 24 h to allow for the infection to develop and either a bolus of RMP (dissolved in methanol) or blank methanol was injected directly into the muscle tissue and incubated for an additional 48 h. PMMA-CD beads were removed from the muscle tissue after this time and CFU counts were performed on both the PMMA-CD bead (via sonication) and the surrounding tissue (via homogenization).

Figure 10 displays the CFU counts remaining on either the PMMA-CD bead (with and without CD microparticles) and the muscle tissue surrounding each bead after treatment with RMP as a percentage of colonies remaining relative to no treatment (i.e., blank methanol). Results, from Figure 10, demonstrated that regardless of the presence of CD microparticles in the PMMA, RMP was able to eradicate a substantial amount of bacteria present in both the tissue and in the biofilm on the surface of the PMMA-CD bead. Specifically, upon treatment with RMP only 0.003% and 0.0009% of bacteria remained on the surface of the PMMA-CD bead and in the muscle tissue, respectively, whereas 0.012% and 0.0026% of bacteria remained on the plain PMMA bead and in the surrounding muscle tissue. Upon exploring the remaining bacterial load after RMP treatment, the refilled PMMA-CD beads were also evaluated for their duration of antimicrobial activity against *S. aureus* in a persistence zone of inhibition study (Figure 11), as a reflection of their capacity to be refilled in the presence of a bacterial biofilm. PMMA-CD beads refilled in the presence of infection were able to clear bacteria for 40+ days, whereas plain PMMA beads (without CD microparticles) were only able to clear bacteria for <7 days.

## 4. Discussion

The aim of this work was to evaluate the possibility of allowing for PMMA-CD composite (containing 15 wt% CD microparticles) to be refilled with antibiotic (RMP) following implantation in bone and soft tissue for the treatment of chronic arthroplasty PJIs. Previous work has demonstrated the ability of PMMA-CD composite to be refilled in an agarose-based tissue phantom, but this model did not sufficiently replicate the properties of diffusion of drug through hard tissue, such as bone. To allow for appreciable antibiotic refilling to occur in the PMMA-CD composite, it is necessary that the PMMA-CD be in contact with a high concentration of antibiotic for a period of time (i.e., that the drug is not immediately cleared away). Since antibiotic cannot freely diffuse through bone, for the bone refilling model it was necessary to penetrate the bone and inject the drug directly into the bone marrow using IO infusion to get a high enough concentration of antibiotic in direct contact with the implanted PMMA-CD.

Collectively, from the results of Figure 3 and Figure 4, it was evident that in order to achieve antibiotic refilling in PMMA implanted in bone, incorporation of CD microparticles was critical, since the depth of RMP diffusion into PMMA-CD composites was significantly greater than PMMA without CD using IO infusion. While the depth of RMP refilling on the periphery of the PMMA-CD composites was mostly uniform on individual samples in the femur refilling model, refilling depth did vary somewhat among samples. Lack of uniformity of RMP refilling depth was generally attributed to heterogeneity in the distribution of CD microparticles in the PMMA-CD composite. For example, if greater amounts of CD microparticles were clustered at the periphery, a greater depth of RMP diffusion would theoretically result. Variation in the distribution of CD microparticles in PMMA could potentially be improved upon by exploration of different mixing techniques of PMMA to enhance the homogeneity of CD microparticles.

The extended duration of antimicrobial activity of the PMMA-CD composite implanted in bone refilled with IO infusion demonstrates the potential of the refilling model to provide the patient with an alternative treatment to PJIs where it may not be necessary to physically remove the infected arthroplasty implant, but simply locally administer more antibiotic as needed. Currently, if a PJI develops, the first line of treatment is systemic antibiotics followed by one- or two-stage revision procedures that result in significant trauma, mobility restrictions, and cost to the patient [33,34]. The ability to refill PMMA-CD composite after it has been implanted in bone to treat prolonged PJIs could help to dramatically improve PJI treatment and patient outcomes without the need for revision surgeries. Furthermore, the PMMA-CD composite system has the potential to be patient-customizable where the antibiotic administered can be tailored to target a specific causative pathogen.

Antimicrobial activity resulting from the plain PMMA refilled in the femur model was attributed to the small amount of RMP that passively diffused into the periphery of the PMMA, whereas the prolonged duration of antimicrobial activity that resulted from PMMA-CD composite was attributed to the greater amount of drug that was able to be refilled (related to the increased depth of diffusion in Figure 4) as well as the more complex affinity-based drug release mechanism of CD. Rather than just passively diffusing out of the plain PMMA, incorporation of CD into the PMMA enabled the drug to bounce in and out of CD’s slightly hydrophobic inner “pockets” due to its affinity for CD prior to diffusing out of the PMMA, thus enabling a more prolonged and consistent release of drug from the PMMA and subsequent antimicrobial activity. Release of RMP from refilled PMMA-CD was not evaluated in this work as our past work has shown that while in direct contact with bacteria, RMP-filled PMMA-CD demonstrated a sustained therapeutic antimicrobial effect. In solution, RMP has exhibited little to no detectable elution from PMMA-CD when released into phosphate buffered saline and other slightly more hydrophobic sinks such as Tween [5]. It is important to note that the femur refilling model was designed under the assumption that the PJI will predominantly develop internally (i.e., osteomyelitis) rather than externally to the bone [35,36]. In the case that the PJI develops externally to the bone, the muscle tissue refilling model may be more amenable.

In terms of the soft tissue refilling model, given that the duration of antimicrobial activity resulting from the refilled plain PMMA beads did not increase after multiple drug refilling cycles, it was hypothesized that there was a threshold for the amount of antibiotic that could passively diffuse into plain PMMA that was reached after a single cycle of refilling with RMP. The increased duration of antimicrobial activity, resulting from additional cycles of antibiotic refilling, demonstrated that with CD microparticles the threshold for RMP refilling of PMMA was substantially higher than that of plain PMMA and that the threshold was not reached after one or two cycles of antibiotic refilling. As PJIs can be prolonged and present between >1 month to more than 2 years following the initial surgery, development of a refillable, patient-customizable PMMA antibiotic delivery system that can be refilled with antibiotics through muscle tissue provides a welcome alternative to many existing, more traumatic, therapies [37].

When composite beads in the soft tissue model were refilled in the presence of a bacterial biofilm, there was a small decrease in the bacterial load of the PMMA-CD composite samples compared to plain PMMA. Nevertheless, it was important to consider that both types of PMMA beads resulted in significant decreases in bacterial load relative to controls without RMP treatment (>99.975% initial bacteria eradicated after RMP treatment). While the bacterial load present on PMMA-CD beads and in surrounding muscle tissue after treatment with RMP was similar in PMMA with and without CD microparticles, composition of PMMA did have a dramatic impact on the amount of RMP filled into PMMA beads in the presence of infection (see Figure 11). As refilled PMMA-CD beads demonstrated a substantially longer duration of antimicrobial activity relative to plain refilled PMMA, this demonstrated that with PMMA-CD beads it was possible to refill antibiotics in the implanted bead and obtain a prolonged duration of antimicrobial activity. The slightly decreased bacterial load in Figure 10 with PMMA-CD composite supported evidence from Figure 11 that a greater amount of antibiotic was filled in the PMMA-CD bead, enabling a longer release to more effectively clear the infection, relative to plain PMMA. These results provided further support regarding the potential therapeutic efficacy of PMMA-CD composite beads to treat PJIs in the surrounding muscle tissue, enabling prolonged antimicrobial activity on demand, even in the presence of a bacterial biofilm. To elaborate, PMMA-CD beads have the ability to be refilled with drug on a patient-specific basis where the type, amount, and frequency of antibiotic bolus can be tailored based upon the causative pathogen to obtain the desired duration of antimicrobial activity.

The large error bars present in Figure 11 were a reflection of the inherent variability of biofilm formation on the surface of PMMA-CD beads between samples. Distribution of CD microparticles and surface roughness were likely some of the factors that contributed to this variability of biofilm formation across samples as roughness dictates the density of bacterial biomass that can adhere to the surface of the PMMA-CD [38,39,40]. The structure and density of the bacterial load in each biofilm can impact downstream properties of the PMMA such as the drug refilling capacity and the rate of drug release, thereby introducing variance into the duration of antimicrobial activity (i.e., Figure 11). This variation in biofilm formation could potentially be improved upon via exploration of different mixing or preparation methods of PMMA-CD to ensure homogeneous distribution of CD microparticles across samples.

## 5. Conclusions

Conventional treatments for PJIs that have developed in the bone and muscle tissue surrounding the prosthetic implant involve radical measures such as months of systemic antibiotics and removal and replacement of the prosthetic, rendering the patient immobile for an extended period of time. The presented work demonstrated the ability of a PMMA-CD composite material to be refilled with antibiotics after being implanted in bone or muscle tissue. Refilled PMMA-CD had an extended duration of antimicrobial activity from 14–40+ days and PMMA-CD composites retained the ability to be refilled with antibiotics even in the presence of a biofilm and tissue infection. Of critical importance, this work demonstrated that PMMA-CD had the ability to provide sustained and on-demand antimicrobial therapy without the necessity to remove the implant if an infection develops. Furthermore, through this work, IO infusion appeared to be a viable, clinically used technique to enable refilling of PMMA-CD after implantation in bone, reporting for the first time the ability to refill PMMA in bone. In summary, PMMA-CD composite is a promising material that is versatile and can be applied to treat two aspects of PJIs—internal osteomyelitis and surrounding muscle tissue infections that can be refilled with antibiotics in a patient-customizable manner regardless of if it is implanted in bone or muscle tissue.

## Figures and Tables

**Figure 1 jfb-12-00008-f001:**
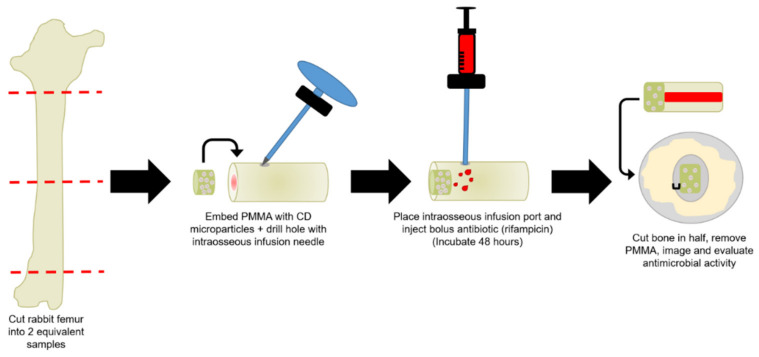
Schematic of the setup of femur poly(methyl methacrylate) (PMMA)-cyclodextrin-(CD) composite antibiotic refilling model where PMMA containing CD microparticles was embedded in the bone and an intraosseous (IO) infusion needle was placed near the implanted PMMA-CD to serve as a port to inject antibiotics. Extent of antibiotic refilling of PMMA-CD using the model was evaluated through cutting the bone in half, explanting the PMMA-CD composite, imaging, and evaluating the duration of antimicrobial activity against *S. aureus*. Each experimental condition was evaluated in quadruplicate using this model.

**Figure 2 jfb-12-00008-f002:**
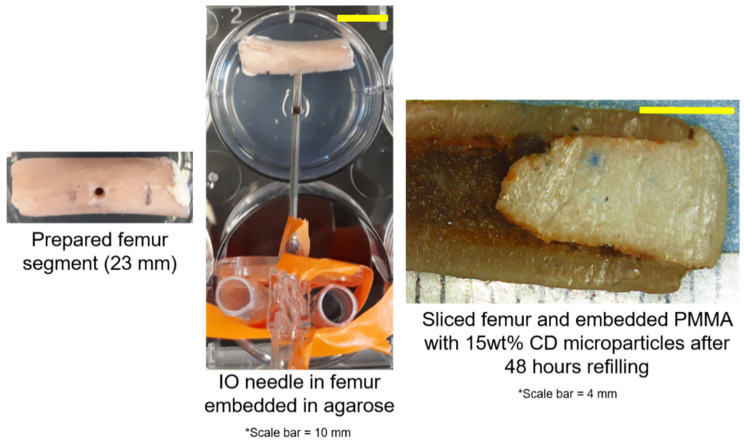
Images of prepared femur segment with embedded PMMA-CD composite (**left**), femur segment embedded in agarose with IO needle inserted (**middle**), and interior of the femur segment 48 h following refilling with rifampicin (RMP) (**right**). Red/orange color along the periphery of the interior of the PMMA-CD was indicative of RMP refilling. Each experimental condition was evaluated in triplicate using this model.

**Figure 3 jfb-12-00008-f003:**
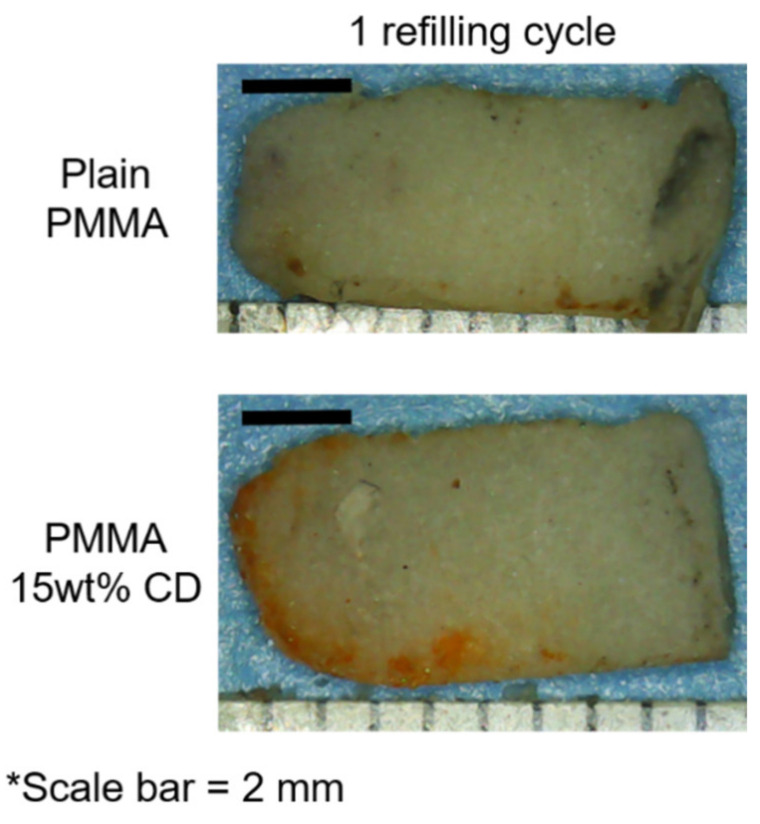
Representative stereomicroscope images of the interior of PMMA-CD composite with and without (i.e., plain) 15 wt% CD microparticles after 48 h of refilling with RMP in femur model. Each experimental condition was evaluated in quadruplicate using this model with a minimum of eight measurements per sample.

**Figure 4 jfb-12-00008-f004:**
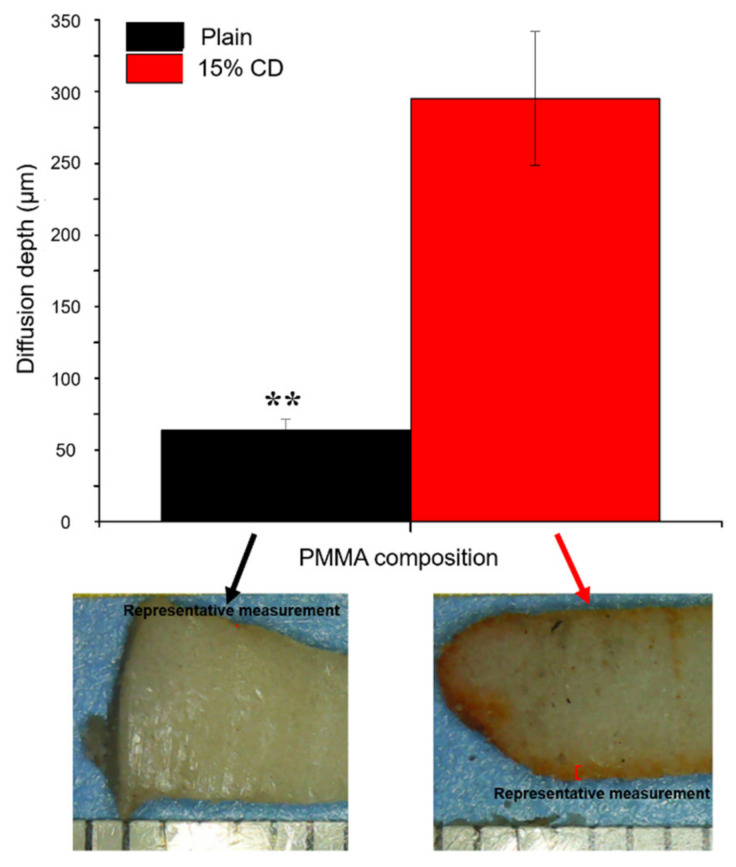
Quantification of the depth or distance of diffusion of RMP into PMMA-CD composite with and without CD microparticles in the femur model and images depicting how representative measurements were collected. Red lines on photos indicate where diffusion was measured from. Each experimental condition was evaluated in quadruplicate using this model with a minimum of eight measurements per sample. ** Indicates *p* < 0.01. Ruler marks on images = 1 mm.

**Figure 5 jfb-12-00008-f005:**
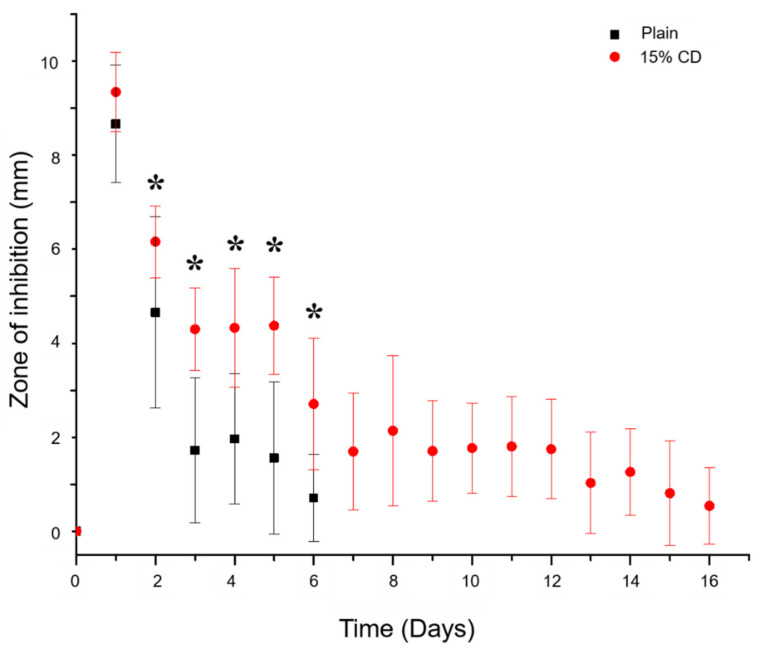
Persistence zone of inhibition study of PMMA-CD composite (with and without 15 wt% CD microparticles) refilled with RMP in femur model against *S. aureus*. Each experimental condition was evaluated in quadruplicate using this model with four measurements per sample. * Indicates *p* < 0.05.

**Figure 6 jfb-12-00008-f006:**
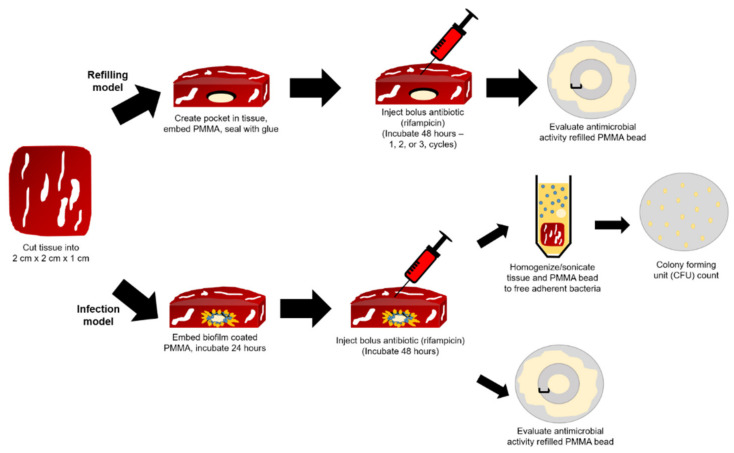
Preparation of muscle tissue sample for implantation of PMMA-CD bead for antibiotic refilling and infection models. Success of antibiotic refilling of PMMA-CD bead through muscle tissue (with and without infection) was analyzed through persistence zone of inhibition studies and colony forming unit (CFU) counts. Each experimental condition was evaluated in triplicate using this model.

**Figure 7 jfb-12-00008-f007:**
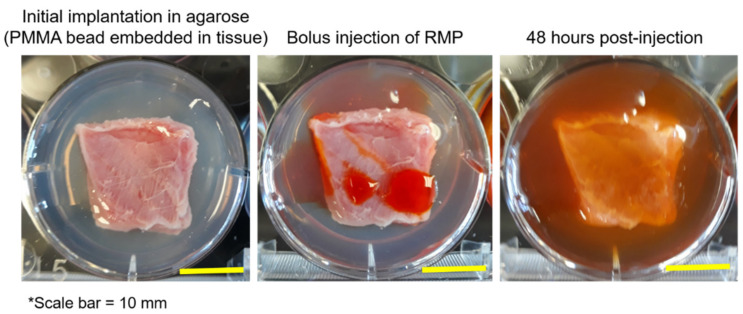
Images of implantation of PMMA-CD bead in muscle tissue in agarose, initial injection of RMP into muscle tissue, and muscle tissue following 48 h of incubation and refilling with RMP. Each experimental condition was evaluated in triplicate using this model.

**Figure 8 jfb-12-00008-f008:**
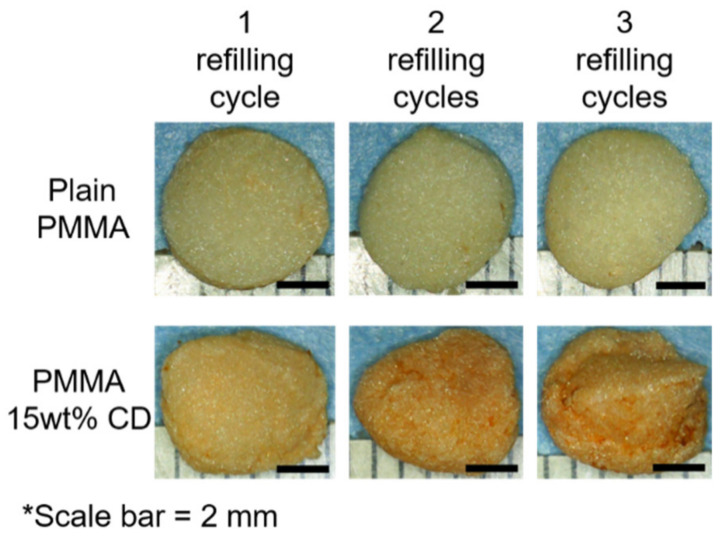
Stereomicroscopy images of PMMA-CD beads after 1–3 cycles of RMP refilling in muscle tissue. Each experimental condition was evaluated in triplicate using this model.

**Figure 9 jfb-12-00008-f009:**
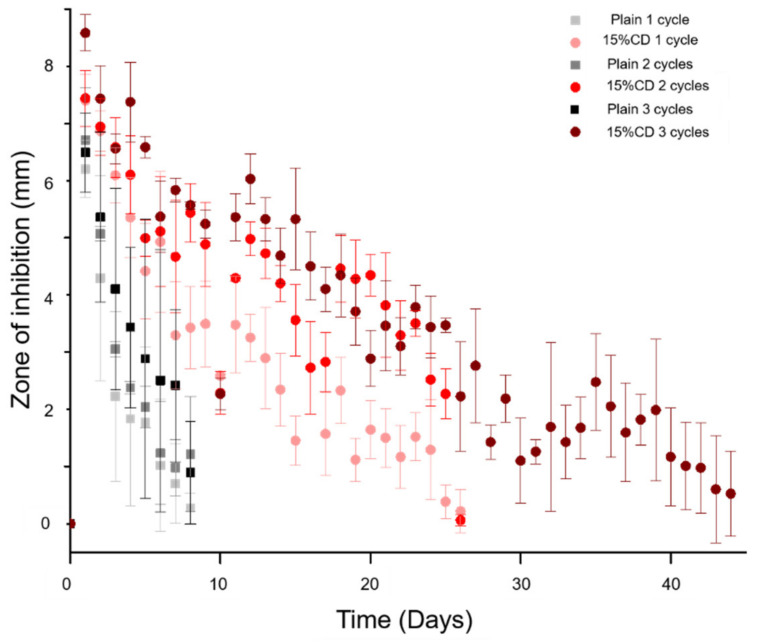
Persistence zone of inhibition study of PMMA-CD beads after 1–3 cycles of RMP refilling in muscle tissue against *S. aureus*. Each experimental condition was evaluated in triplicate using this model with four measurements per sample.

**Figure 10 jfb-12-00008-f010:**
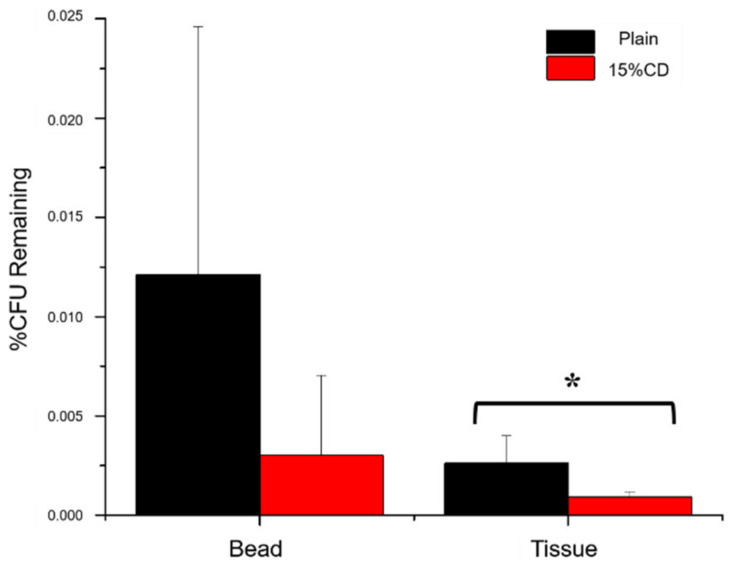
Percent remaining CFU counts of *S. aureus* on PMMA-CD beads (with and without 15 wt% CD microparticles) and respective surrounding muscle tissue after treatment with RMP relative to no treatment. Each experimental condition was evaluated in triplicate using this model. * Indicates *p* < 0.05.

**Figure 11 jfb-12-00008-f011:**
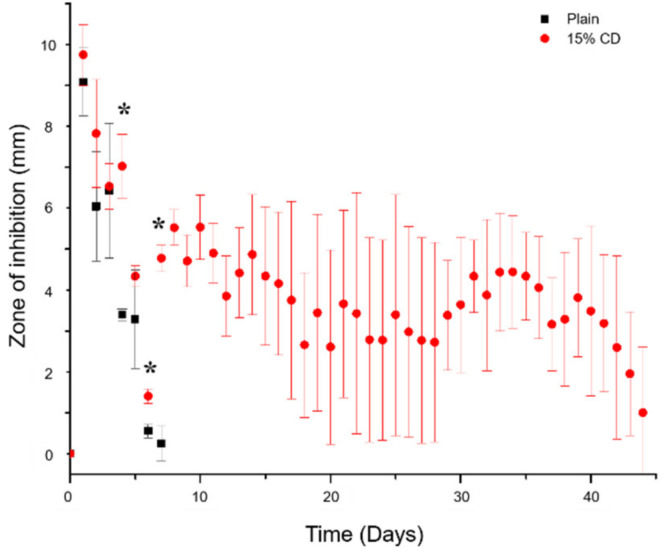
Persistence zone of inhibition study of PMMA-CD beads explanted from muscle tissue infection model after being refilled with RMP for 48 h. Each experimental condition was evaluated in triplicate using this model with four measurements per sample. * Indicates *p* < 0.05.

## Data Availability

Data sharing not applicable.

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
