# Peer review of "Poly(methyl methacrylate) Bone Cement Composite Can Be Refilled with Antibiotics after Implantation in Femur or Soft Tissue"

_jfb, 2021, doi:10.3390/jfb12010008_

Round 1

Reviewer 1 Report

The authors explored the possibility of refilling PMMA-CD composite beads through muscle tissue if they are initially covered in bacterial biofilm, a modified soft-tissue refilling model being developed where biofilms were formed on composite beads prior to implantation in the tissue. This article shows interesting results, it is well-written and I recommend its publication in Journal of Functional Biomaterials.

  • Please indicate the number of replicates in the figure captions.
  • Regarding the characterization of beta-cyclodextrins microparticles and PMMA-CD composite, please give more details. Did you publish the characterization before? 

Reviewer 2 Report

The manuscript "PMMA bone cement composite can be refilled with antibiotics 2 after implantation in femur or soft tissue" it is very interesting from scientific and medical point of view.

The work describes a interesting as a way to use the ability of a PMMA composite material to be refilled with antibiotics after being implanted in bone or muscle tissuel. 

Thereis description of the protocol which would enable anyone else to replicate the experiment.

The manuscript "PMMA bone cement composite can be refilled with antibiotics 2 after implantation in femur or soft tissue" it is very interesting from scientific and medical point of view.

The work describes a interesting as a way to use the ability of a PMMA composite material to be refilled with antibiotics after being implanted in bone or muscle tissuel. 

Thereis description of the protocol which would enable anyone else to replicate the experiment.

The manuscript "PMMA bone cement composite can be refilled with antibiotics 2 after implantation in femur or soft tissue" it is very interesting from scientific and medical point of view.

The work describes a interesting as a way to use the ability of a PMMA composite material to be refilled with antibiotics after being implanted in bone or muscle tissuel. 

Thereis description of the protocol which would enable anyone else to replicate the experiment.
